# Underground Microseismic Event Monitoring and Localization within Sensor Networks

**DOI:** 10.3390/s21082830

**Published:** 2021-04-17

**Authors:** Sili Wang, Mark P. Panning, Steven D. Vance, Wenzhan Song

**Affiliations:** 1College of Engineering, University of Georgia, Boyd Graduate Building 128, Athens, GA 30602, USA; sw84815@uga.edu; 2Jet Propulsion Laboratory, California Institute of Technology, Pasadena, CA 91109, USA; mark.p.panning@jpl.nasa.gov (M.P.P.); steven.d.vance@jpl.nasa.gov (S.D.V.)

**Keywords:** seismic source localization, Gaussian beam migration, time reverse imaging, distributed sensor network, real-time seismic simulation

## Abstract

Locating underground microseismic events is important for monitoring subsurface activity and understanding the planetary subsurface evolution. Due to bandwidth limitations, especially in applications involving planetarily-distributed sensor networks, networks should be designed to perform the localization algorithm in-situ, so that only the source location information needs to be sent out, not the raw data. In this paper, we propose a decentralized Gaussian beam time-reverse imaging (GB-TRI) algorithm that can be incorporated to the distributed sensors to detect and locate underground microseismic events with reduced usage of computational resources and communication bandwidth of the network. After the in-situ distributed computation, the final real-time location result is generated and delivered. We used a real-time simulation platform to test the performance of the system. We also evaluated the stability and accuracy of our proposed GB-TRI localization algorithm using extensive experiments and tests.

## 1. Introduction

Monitoring the underground seismic events of a planet is an important part of planetary science, as it helps to explain the subsurface structure of the planet. Current planetary exploration techniques have successfully gathered seismic data from Mars. In 2018, the InSight Mission to Mars started to collect seismic data, by using a single seismic station to clearly record marsquakes for the first time and begin constraining subsurface structure from the crust to the core [1]; in 2024, NASA’s Europa Lander mission concept payload included a seismic package for characterizing the local environment of Europa’s ice and possibly the characteristics of its ocean and deeper interior [2]. The planned Dragonfly mission concept to explore Saturn’s moon Titan also includes a seismic package. While each of these missions has a single-station approach, future missions to may include multiple seismic nodes. The Lunar Geophysical Network is a prominent example of a possible near-future planetary network. The detection and localization of seismic events is the foundation of the deeper exploration of the planet. Although passive seismic source monitoring is a mature technology on Earth, it is difficult to implement on other planets or moons. The challenges include, but are not limited to:The installation of the sensors being difficult, and the hardware system is perishable and may not be retrieved.The surface of the target planet or moon has topographic relief; usually we need to account for topographic variation.The communication bandwidth between the planet, satellite and Earth is limited.

The localization algorithm we use is Gaussian beam time-reverse imaging (GB-TRI). Time-reverse imaging (TRI) is a passive source imaging method [3] that uses time series data gathered by a surface sensor network to generate a source energy map [4]. The possibility of implementing TRI in distributed format was first discussed in 2015 [5], and first realized in 2018 [6]. GB-TRI uses a Gaussian beam to approximate the time domain TRI; it was first used in real-time underground activity localization in 2019 [7]. Compared with the time domain TRI, GB-TRI is easier to calculate in a distributed and decentralized structure [8]. In this work, we optimize GB-TRI for the influences of topographic variation, high packet loss rate and resource limitations. Our experimental result shows that the optimized GB-TRI works well on the proposed system.

To evaluate our system and algorithm, we conducted both one-time experiments and real-time experiments. In particular, we designed an AWS (Amazon Web Services)-based real-time seismic simulation platform. The platform includes InfluxDB, a database designed for time series data analysis [9]. With this platform, we can generate the events in real-time, and show the seismic data and the localization result as an animation. Since the field deployment of our system is expensive, this tool can help us test the algorithm in an affordable way. We evaluated our system and algorithm with simulation experiments. The results show that our system and algorithm work well to localize the underground seismic events while maintaining good performance.

The contributions of our work mainly include three achievements:This is the first time an end-to-end distributed system framework for passive seismic source localization has been designed.An optimized Gaussian beam TRI algorithm is proposed to solve the passive source locating problem in a distributed way using a clustering structure. Compared to the former algorithm, it is a robust and lightweight algorithm to implemented on smart sensors with limited computational resources.We developed a simulation and experimental platform with networking emulation, synthetic data generation and visualization capabilities. This is the first work to show a real-time microseismic localization result.

The rest of the paper is organized as follows. Section 2 presents the related work. Section 3 provides information of algorithms about the microseismic event localization end-to-end process and the distributed system design. In Section 4, we report on experiments with synthetic seismic data. Section 5 presents the discussion about this work. The conclusion and future work are presented in Section 6.

## 2. Related Works

In the literature, there are several passive source localization algorithms: arrival time difference [10,11], cross-correlation [12], range difference [13] and migration [14]. In recent years, the time-reverse imaging method has become a popular method for passive source localization for its accuracy and anti-noise performance. The imaging algorithm uses time series data gathered by sensor networks to generate a source energy map and shows the probable location of underground activity.

The traditional reverse-time migration (RTM) algorithm in radar systems uses an active source signal and the received signal to detect the underground scattering to form a structural image; it is widely used in structural health monitoring [15]. TRI generates imaging results without cross-correlation [8]. To generate a source energy image, other image conditions such as stacked, maximum or geometric mean are used [14]. In a sensor network, TRI calculates the backward-propagated wavefields in each single-sensor node individually, which is naturally distributed [5]. Conventional time domain TRI uses a finite-difference time domain (FDTD) algorithm to solve the wave equation [16]. The computational cost of FDTD depends on the number of iterations of the discrete wave equation calculation and the size of spatial grid boundary condition [17]. If the grid size is large, running the TRI method in each embedded sensor node is energetically expensive. Moreover, when the TRI algorithm is implemented in a distributed sensor network, the communication cost increases with the number of sensors [18].

Gaussian beam migration (GBM) is a high frequency approximation of FDTD migration [19], popular for its flexibility and efficiency [20]. GBM is especially suited for sparse data sets [21]. Both the acoustic and elastic wave equations can be solved with the GBM method [22,23,24]. Since TRI is also based on the wave equation solution, there have been prior efforts using GBM to solve the TRI problem. Popov has combined reverse-time migration with Gaussian beams to solve the problem and also introduce an improved imaging condition suitable for computation of velocity contrasts at reflectors in the framework of the true-amplitude concept [24,25]. A green function-based approach to combine the RTM and GBM was proposed in 2014 [23]. In 2017, the reverse-time migration with elastodynamic Gaussian beams method was proposed [26].

Recent work has implemented the traditional geophysical algorithm on a distributed network system. In 2015, Sun tried to implement the TRI algorithm in a distribute system [5]. A ambient noise seismic imaging algorithm was implemented in a real-time distributed system by Valero and optimized by Wang [27,28]. Song tried to run the geophysical algorithm on a distributed sensor network to make a subsurface camera [29]. As a whole, distributed subsurface imaging algorithm is an active research topic in this decade.

As far as we know, our work is the first which achieved a second-order, real-time microseismic localization result. A comparison of different microseismic localization methods is listed in Table 1.

## 3. Algorithm and System Design

### 3.1. Background and System Design

Usually, seismic data are collected by three-component seismometers deployed on the surface. Figure 1 illustrates a typical seismic source monitoring system with geophone arrays [7]. The localization problem can be calculated in both 2D and 3D. Typically, the sensors are deployed along the measuring line for 2D models or arranged as a 2D array for 3D models. To simplify the calculation, usually, the localization algorithm is calculated in 2D first and then extended to 3D with interpolation methods.

Figure 1 shows the propagation of the seismic wave and how the surface sensor network detects this wave. The wave will propagate following the wave equation. For the different nodes, the arrival of the same wavelet should differ in time. We use this time difference between nodes to calculate the source location using the backward wave propagation algorithm.

In planetary exploration projects, the deployment of the sensors is expensive and difficult. For that reason, we only consider the 2D model that only requires sensors deployed on one measuring line. The system comprises several sensor clusters. The clusters are connected to each other by cable. Figure 2a shows the schematic diagram of the sensor system: each blue triangle represents one cluster, which includes several independent sensor nodes; the clusters are connected physically by a strong cable, and can be dropped from a low-altitude spacecraft. After the sensor clusters land on the surface, they will naturally form a measuring line. However, the landing location of the clusters is not a perfect straight line and the final position of the sensor is affected by topographic variation, as Figure 2b shows.

The collected data are visualized using InfluxDB and Grafana. InfluxDB is a database designed for time series data analysis [30]. We store the time series data in the InfluxDB databases, and the Grafana visualization tools can directly use the InfluxDB databases to show the waveform in real-time [31].

The structure of the system is designed for the distributed GB-TRI algorithm, which distributes the calculation evenly to the individual nodes. With the distributed GB-TRI algorithm, the work load is balanced in the whole system, allowing other nodes to compensate the loss of any single node. The distributed calculation makes the system more stable in the challenging working environment. The clustering structure is also beneficial to the communication load. We need two inputs to solve the Gaussian beam approximation in the backward wavefield: the recorded seismic data and the initial angle of the Gaussian beam. When the signal–noise ratio (SNR) is low, we use data from neighboring nodes to solve for the initial angle. With this clustering structure, we avoid the long distance transmission of the raw data, and hence reduce the communication cost. The detail of the algorithm is discussed in the next section.

### 3.2. Distributed Gaussian Beam Time-Reverse Imaging Algorithm

Passive source imaging is a good approach with which to realize micro seismic source location. The imaging algorithm uses time series data gathered by the surface sensor network to generate a source energy map showing the probable location of underground activity. To get the location of the source, we construct the backward wavefield with the observed data gathered by the surface sensors. The final result is an energy map representing the energy distribution of stacked backward wavefield.

TRI is a passive source imaging method [3], which uses time series data gathered by surface sensor network to generate a source energy map [4]. To detect each of these activities separately and continuously, we separate the continuum data into segments with short time windows. The window size is chosen carefully to make sure that only a few major activities have occurred in each segment. After generating a source energy map, we refresh the image with the data from next time window. TRI generates source images by solving the backward wave equation. FDTD and a perfect matched layer absorbing boundary conditions are used in the traditional TRI algorithm. The output image is an energy map representing the concentrated source energy in both time and space. On this map, we can see bright spots occurring on the location of underground activities.

Equation (Equation 1) is a typical TRI wave equation, involving spatial coordinates (z,x) and replacing time coordinate *t* with −t.
(1)$$\frac{1}{v^{2}\left(z,x\right)}\frac{\partial^{2}u_{Bt}\left(z,x,t\right)}{\partial {\left(-t\right)}^{2}}=-{▽}^{2}u_{Bt}\left(z,x,t\right)$$
$u_{Bt}$ is the backward wavefield; v(z,x) is the velocity corresponding to the point (z,x) on the velocity model. To solve this equation, we need an initial value acting as a boundary condition. Instead of using the free surface boundary condition, the TRI method defines the surface boundary condition with the value of the wavefield on the surface [32]:(2)$$u_{Bt}\left(z=0,x,t\right)=u_{Ft}\left(z=0,x,T-t\right),$$
where $u_{Bt}$ is the backward wavefield at time *t* and $u_{Ft}$ is the forward wavefield at time T−t, which is the data recorded at T−t by receivers. When we use the gathered seismic data at z=0 as the boundary condition and process from time t=T to time t=0, we can get a time-reversed backward propagated wavefield; *T* is the length of the time window. [33] This process is equivalent to stacking individual backward wavefields generated by one single trace of seismic data.

The time domain TRI imaging condition is shown in Equation (Equation 3). The stacked wavefield of all of the receivers represents the energy distribution of the backward wavefield toward time. The TRI source energy map is coordinated with the stacked value of the wavefield on the time scale.
(3)$$I\left(z,x\right)=\int_{t_{0}}^{t}\left(\sum_{i=1}^{N}U_{i}\left(z,x,t\right)\right)$$
where $U_{i}$ indicates the backward wavefield of the *i*th single receiver. This imaging condition is not robust due to the energy decrements. A hybrid image condition is proposed to get high resolution image when the signal–noise ratio (SNR) is low [5], a mix condition of stacked imaging and cross-correlation imaging. Considering thaht the Gaussian beam wavefield solution is solved in the frequency domain with ray direction coordinates and has less energy decrements, we use the maximum wavefield value to replace the stacked wavefield value, shown as Equation (Equation 4). The physical meaning is that the maximum energy occurs at the specific time period and the target point (s,n).
(4)$$I\left(s,n\right)=max(ifft\left(\sum_{i=1}^{N}U_{i}\left(s,n,w\right)\right)).$$

If the input data contain the signal of a microseismic source, the backward wavefield at $\left(z_{0},x_{0}\right)$ has its maximum value when the energy of all of the individual wavefields is concentrated. We define this value as the imaging value on the energy map. It is easy to see that the maximum value on this map occurs at the location of the source. The limitation of traditional FDTD-based TRI is that a sudden change of velocity and the artificial absorption boundary condition may cause dispersion, which aliases the result. Traditional FDTD is hard to implement on a parallel or distributed computing system because it is an iterative method in the time domain. To solve this problem, we combine the Gaussian beam algorithm with the traditional TRI algorithm.

Gaussian beam migration is a ray path method to solve the wave equation. In a two-dimensional TRI problem, Equation (Equation 1) is transformed into a frequency domain scalar wave equation:(5)$$\frac{w^{2}}{v^{2}\left(z,x\right)}U_{B}\left(z,x,w\right)=-{▽}^{2}U_{B}\left(z,x,w\right)$$
where *w* is the frequency, $U_{B}$ is the backward wavefield in the frequency domain and $v^{2}\left(z,x\right)$ is the velocity corresponding to the spatial location. To solve this wave equation, there are three steps: (i) ray tracing; (ii) solving the wavefront curvature functions; (iii) construction of the wavefield [19].

The beam center of the Gaussian beam is generated by ray path equations:(6)$$\frac{dx}{dt}=vsin\delta ,\frac{dz}{dt}=vcos\delta ,\frac{d\delta }{dt}=-\frac{dv}{dx}cos\delta +\frac{dv}{dz}sin\delta$$

The beginning conditions are $t=t_{0}$, $\delta =\delta_{0}$, $x=x_{0}$ and $z=z_{0}$. $t_{0}$ is the start time; $\delta_{0}$ is the included angle between the ray path and z axis; $\left(z_{0},x_{0}\right)$ is the location of the receiver. The ray-centered coordinates *s* and *n* are generated by Equation (Equation 6) and ds=vdt.

The width and front of the Gaussian beams depend on the wavefront curvature functions P(s) and Q(s). Both of them are complex functions. Using a velocity model, V(s,n), P(s) and Q(s) can be solved by Equations (Equation 7) and (Equation 8):(7)$$\frac{dQ(s)}{ds}=v\left(s\right)P\left(s\right)$$
(8)$$\frac{dP(s)}{ds}=-\frac{1}{v^{2}\left(s\right)}\frac{\partial^{2}V\left(s,n\right)}{\partial n^{2}}Q\left(s\right)$$

The initial values $P_{0}$ and $Q_{0}$ determines the initial width and wavefront curvature of the Gaussian beams. With these two functions, the frequency domain wavefield under Gaussian beam ray-centered coordinates can be calculated as an asymptotic solution of Equation (Equation 5), defined as Equation (Equation 9).
(9)$$U\left(s,n,w\right)=U_{0}{\left[\frac{v(s)}{Q(s)}\right]}^{1/2}exp\left[iwt\left(s\right)+\frac{iw}{2}\frac{P(s)}{Q(s)}n^{2}\right]$$
where *w* is the frequency, $U_{0}$ is the frequency domain wavefield value on (s=0,n=0), U(s,n,w) is the backward wavefield corresponding to (s,n,w) and v(s) is the velocity at the beam center corresponding to *s* [34].

The backward wavefield can be represented by a collection of Gaussian beams with specified parameters. The initials values $P_{0}=P\left(s_{0}\right)$ and $Q_{0}=Q\left(s_{0}\right)$ are under the constraint that $P_{0}/Q_{0}$ is always an imaginary. Usually, we use $P_{0}=i/V_{0}$ and $Q_{0}=w_{r}w_{0}^{2}/V_{0}$ as initial values. $V_{0}$ is the velocity at the point (s=0,n=0); $w_{r}$ is a reference frequency based on the frequency range of the seismic wavelet; and $w_{0}$ depends on $w_{r}$. A good choice of $w_{0}$ is $2\pi V_{a}/w_{r}$, where $V_{a}$ is the average velocity. The advantage of the Gaussian beam wavefield solution is that the wavefield can be solved independently with coordinates and beam parameters, which makes the algorithm suitable to be distributed. It is also an algorithm that can reduce the effect of the fracture and break in the velocity model. In the application of the TRI source location, considering that the exact velocity model is difficult to generate in real-time field tests, the ray-center coordinates generated by ray tracing might be misleading. To reduce this effect, the Gaussian beam with ray tracing can be simplified to a Gaussian beam with a straight ray direction coordinate shown in Figure 3. In this way, the ray path coordinates *s* and *n* can be solved directly without Equation (Equation 6); the complex functions P(s) and Q(s) can be solved using:(10)$$Q\left(s\right)=\int_{s_{0}}^{s}v\left(s\right)P\left(s\right)ds$$
(11)$$P\left(s\right)=\int_{s_{0}}^{s}-\frac{1}{v^{2}\left(s\right)}\frac{\partial^{2}v\left(s,n\right)}{\partial n^{2}}Q\left(s\right)ds$$

If a seismic event is detected in the former step, we start to calculate the backward wavefield and generate the energy map. As an asymptotic wavefield solution, we can construct a Gaussian beam backward wavefield similarly to the TRI backward wavefield. However, only the active area where the seismic event occurred is desired. In this case, only the wavefield in the area of one Gaussian beam is calculated to generate a source map. With the ray direction coordinates and Gaussian beam wavefield solution, we can calculate the wavefield parallel. Additionally, we avoid the absorbing boundary condition and the scattering effect of the velocity model. The Gaussian beam requires a smooth velocity model, which means it is not necessary to use an accurate velocity model as input. This attribute makes GB-TRI especially suitable for passive source localization when the velocity model is unknown.

### 3.3. Calculate the Initial Angle with the First Arrival Peak

To construct the Gaussian beam backward wave field, in addition to the observed wave field data, we also need the initial wave propagation angle. The three-component seismometer can obtain wave propagation angles with one node independently. Figure 4a shows the case of 2D elastic wave recording; the received data has two orthogonal ground-motion records corresponding to the horizontal and vertical components (respectively noted X and Z). Configuration of a 2×2 covariance matrix over the two components is:(12)$$C=\left[\begin{array}{cc} cov(X,X) & cov(X,Z)\\ cov(Z,X) & cov(Z,Z) \end{array}\right]$$
where cov(•) denotes the covariance. Eigenvectors of Equation (Equation 12) form an orthogonal base, from which the wave orientation can be estimated. We organize eigenvalues so that λ1 is bigger than λ2 with Ui,i=1,2 as the eigenvectors. Then, the desired angle θ is the direction of U1. Practically, these eigenvalues and eigenvectors can be solved by singular value decomposition (SVD) as the red line in Figure 4b.

The seismic waveform contains both P and S waves. Usually the P wave velocity is larger than the S wave velocity, so the first arrival peak of the wave should always be the P wave. P waves are also known as compressional waves. When subjected to a P wave, particles move in the same direction that the the wave is moving in, which is the direction that the energy is traveling in. For that reason, by solving the movement direction of first arrival, we get the direction of wave propagation.

Considering that our installation of the sensor is not perfect, there may be significant noise in the seismic observation. When the SNR is low, we need to use an alternative method to calculate the initial angle. One solution is applying the traditional TRI method with the raw data collected by the neighboring nodes inside the cluster. As the nodes in the same cluster are close to each other, the communication cost is not significant. However, the short distance between nodes also limited the effective area of the TRI method in the superficial layer of the ground. In this case, we refer to this algorithm as superficial layer TRI.

Superficial layer TRI solves for the backward wavefield with the TRI algorithm in a small area. The algorithm is the same as a typical TRI algorithm, but the calculation is limited in the spatial domain. First, every node reads the raw data from its sensors and pre-processes the raw data. The pre-processing includes cutting the continuum data into segments, selecting the first part in each segments with a shorter time window and implementing time-reversal. The window length is not necessarily long enough to cover the whole wave propagating time from source to receiver. A short window that contains the signal of the seismic event is sufficient. Instead of solving the whole wavefield, the algorithm only solves the superficial layer and short offset area. As shown in Figure 5, the output image is a beam toward the source location, and we define the direction from the sensor to the maximum point on TRI beam image as the initial direction of Gaussian beam from this sensor.

### 3.4. Workflow of GB-TRI

A typical GB-TRI imaging workflow for a single node in the distributed sensor network is shown in Figure 6. Most of the signal processing and wavefield calculation steps are completed independently. The data exchange between nodes only occurs before the final imaging process. All nodes have the same work flow, so this is a decentralized algorithm which is robust even if the packet loss rate is high or several nodes have stopped functioning.

The innovation of our algorithm is that the Gaussian beam wavefield extrapolation process can be calculated independently on a spatial axis. In this case, we only calculate the wavefield in the range of the Gaussian beam; this process increases the efficiency of calculation.

Define the time length of raw data as *t*, the grid size of velocity model as z×x, the total number of nodes as *l*, the average number of neighbor nodes as *n* and the average transmission length as *m*. Table 2 shows the communication and computational cost of centralized TRI, distributed TRI and GB-TRI. The centralized algorithm has the lowest computation cost, and the distributed TRI algorithm has the highest computation cost. The communication cost depends on the parameters. In the case that the time is long or the sampling rate is high, GB-TRI has the lowest communication cost.

## 4. Experiments and Evaluations

### 4.1. Real-Time Simulation of the System

To simulate the real-time performance of the proposed system, we designed an end-to-end software system to demonstrate the distributed source localization algorithm, whose flow chart is shown in Figure 7. The system provides real-time seismic signal generation and real-time seismic source localization. The software-based architecture makes it easy to demonstrate and validate the algorithm. Additionally, the cloud-based data structure will make this system easy to connect with hardware-based architectures in future.

The seismic data were generated with a GUI written in Python. We generated the forward wavefield data using the FDTD method modeled based on an input velocity model. When we clicked on the graphical interfaces, an event was set up with a Ricker wavelet, and the location of this event was shown on the screen as a green square. The noise was also considered in the data generation. Figure 8 shows the GUI. In this test when we clicked persistently from left to right. The system generated the seismic vibration data in real-time and sent data to the cloud influx database and displayed it with Grafana. Figure 9 shows the real-time visualization of the seismic data, which was refreshed every 5 s.

The imaging script is another independent Python script. It queries data from the cloud database and calculates the inversion in real-time. In this test, we set the length of the sliding window as 5 s and refreshed the result every 5 s. The output energy map is printed on the screen and refreshed automatically. In this case, the output was a continual animation.

The real-time simulation experiment was conducted on a desktop with 8 GB of memory and a 4-core CPU. Both the forwarding part and the inversion part were running on the same desktop. In this experiment, we generated a sequence of events in 20 min; the locations are shown in Figure 8 as the green squares. We took screenshots every five minutes. Figure 10 shows four screenshots which show the locations of the events we generated with this GUI. The result shows that the seismic events occurred in different locations. With this GUI, we saw the the real-time performance of the system; the localization result is clear and the output can refresh every 5 s with limited use of computational resources.

### 4.2. Algorithm Evaluation with an Offline Experiment

The GB-TRI imaging algorithm has good performance under ideal conditions. However, in the real sensor network, noise, packet loss and asynchronization occur. To evaluate the performance of our algorithm in different environments, we also conducted a series of offline one-time experiments.

#### 4.2.1. Noise Test

An noise experiment was performed on the Marmousi velocity model [35], the geometry of which was based on a profile through the North Quenguela trough in the Cuanza basin, which is a common velocity model used to test algorithms in exploration seismology literature. We selected a square area in the Marmousi model containing multiple layers, as shown in Figure 11. The velocity range was from 1500 m/s to 3500 m/s. The size of velocity model was 300×300; both the depth interval dz and the width interval dx were 4 m. The deployed surface sensor array included 45 sensors from x=160 m to x=1040 m; the interval between sensors was 60 m. We set up a single seismic event at (*z* = 520 m, *x* = 880 m) and generated seismic data with the FDTD method; the time length of data was 0.5 s; the time interval was 0.5 ms.

Figure 12 shows the FDTD-TRI and the GB-TRI imaging results with different noise levels. To generate a GB-TRI energy map, we needed to implement Gaussian blur first on the velocity model to meet the smooth velocity model requirement of GB-TRI algorithm. On the imaging result, we can see the center of the energy map is almost at the true location of seismic source. GB-TRI can get a similar localization result to the FDTD-TRI result shown in the first row. The result shows that our algorithm performs well, even in a complex velocity environment. We discuss the accuracy of the locating result later.

We added white noise at the difference level of the measured SNR. Figure 12b–d shows the dataset with noise and the final energy map at SNR = 10 dB, SNR = 0 dB and SNR = −10 dB, respectively. On the image of the observed seismic dataset, we plotted the waveform of the observed data at x=1000 m. From this waveform, the difference of the noise level is obvious. The result shows that when SNR = 10 dB, the imaging result is almost same as the noise-free result; when SNR = 0 dB, the area near source location does not change too much, but there are some unexpected structures shown in the near surface area; when SNR = −10 db, it is obvious that several Gaussian beams spuriously located energy on the right edge of the energy map, and the spot near the source location has some distortion compared with other energy maps. In general the energy distribution on the energy map is more disperse with a lower SNR. The noise affects the final imaging stage: with the stacked image condition, the SNR of the final energy map result depends on the number of folds. The similarity between energy maps with different SNRs shows that the algorithm is robust in noisy environments. Although a low SNR may cause some unexpected structure, the source energy concentration is still significant around the source location area.

#### 4.2.2. Packet Loss Test

In a sensor network, packet loss is a common phenomenon, which sometimes makes the algorithm fail to generate a good result. To deal with this problem, the algorithm should have a tolerance for packet loss. In the TRI algorithm, the stacked or cross-correlation imaging condition, which relies on folds, has natural resistance to packet loss. Figure 13 shows the output energy map generated by FDTD-TRI and GB-TRI after adding different rates of packet loss. Figure 13a is the energy map without packet loss; Figure 13b,c are energy maps under 40% packet loss and 70% packet loss, respectively. Since the packet loss occurs randomly, the output energy map could vary from one test to another. For example, as we can see in the Figure 13b, some packets from middle sensors were missed. The result was that there were two obvious beams on the energy map, but the energy around the source location was more concentrated than in Figure 13a. Figure 13c shows the opposite condition: the wave field data packet remaining on the energy map is concentrated. In this case, the source location is clear in horizontal and *s* directions, but not well-localized in vertical and *z* directions. Interestingly, the FDTD-TRI result shows some obvious structures in the near surface area under high packet loss rate, but GB-TRI result does not show that structure. The reason could be that the Gaussian beam construction process introduces a smoothing effect on the wave field, which may compensate for packet loss. This test shows that our algorithm performs well even when the percentage of packet loss is high. Even if the packet loss rate is as much as 70%, we can still find localized source energy on the output energy map.

To measure the accuracy of GB-TRI locating result, we used the deviation between the location result and the true location to represent the quality of result. We define the maximum point on the energy map as the location result, and the distance between the location result and the true location is the deviation. Figure 14 shows the comparison of location results and true location under different noise and packet loss conditions. The red circle is the true location of the seismic source, and the blue star is the location result given by the GB-TRI algorithm. As we can see in Figure 14, the deviation is not affected too much by noise or packet loss. This result also shows that our algorithm is robust with respect to both high noise levels and packet loss. To measure the location result in a objective way, we random selected the location of seismic source and ran the experiment independently 20 times under each condition. We used the average of 10 output deviations to validate the accuracy of the locating result. Table 3 shows the average deviations under different SNRs and packet loss rates. The differences between the deviations under different conditions are not significant, which means that the primary cause of error in our algorithm is not noise or packet loss. We also estimated the average localization error of the FDTD-TRI method. The result shows that the FDTD-TRI localization result is very robust; however, the absolute error in the noise-free and no-packet-loss conditions is larger than for the GB-TRI result. The average error of FDTD-TRI is 32.3333 m. Theoretically, the error in GB-TRI is caused by the inaccurate velocity model and the approximate wavefield calculated by the GB wave equation. In the field test, it is hard to get an accurate velocity model. In GB wave field extension, a Gaussian mean filter is applied to the velocity model, which can reduce the effect of local uncertainty. If there is a global offset on the velocity model, the result will also show an offset. However, if the velocity model is relatively stable, we can also find the approximate route of the object. We should notice that even if there are deviations between the true location and the calculated location, the deviation is around 20 m under sustainable noise and packet loss. The location result is precise enough to satisfy the basic resolution requirement in a 1200 × 1200 m velocity model. Considering the efficiency of the algorithm, it is suitable to be implemented in a real-time distributed system.

#### 4.2.3. Time Synchronization Test

Due to the asynchrony of the independent devices, data gathered from different nodes may not be exactly synchronized in time. To test the robustness of the algorithm, we added random delay to our synthetic data. The velocity model we used in this experiment was still the Marmousi model, and the center frequency of the wavelet was 50 Hz. First, we added random delay in the range of 5 ms. The seismic section shown in Figure 15a. Figure 15b,c shows the FDTD-TRI result and GB-TRI result. The results show that the energy of the source still concentrates on the location of the source. This result shows that both FDTD-TRI and GB-TRI algorithms have basic robustness towards asynchronous data. If we add random delay in the range of 25 ms, the seismic section is shown in Figure 15d. Figure 15e,f shows the FDTD-TRI result and GB-TRI result. We found an interesting phenomenon: the GB-TRI result demonstrated better localization than the FDTD-TRI result. Considering that GB-TRI is calculating the wavefield in the frequency domain, it may deal with poorly-synchronized data more naturally.

#### 4.2.4. Undulating Surface Test

The final test was an undulating surface test. To simulate topographic variation, we created a layered velocity model with an undulating surface layer, which is shown in Figure 16. Both the depth interval dz and width interval dx were 5 m. The sensors were deployed along the surface and the source event was located at point (150,150). Since the sensor’s deployment was not horizontal, we needed to use the two-dimensional positions of the sensors in the TRI algorithm. Although the input was changed, the complexity of the algorithm did not change.

Similarly to the previous experiments, we generated the synthetic data with the FDTD method, and the seismic record is plotted in Figure 17a. The results show that the record was distorted because the observing surface was not horizontal. Figure 17b shows the localization result of this experiment. As we can see, the energy was well-concentrated to the location of the source event. From this result, we can conclude that the GB-TRI algorithm can be easily adapted to a setting with topographic variation. Although the algorithm itself is adjustable to topographic variation, it requires the exact 2D position of the sensors, and so obtaining very good estimations of sensor placement and the variation of surface topography are critical to the method.

## 5. Discussion

In the Section 4, a real-time experiment was shown. The experimental result shows that the presented distributed microseismic event localization framework can generate the real-time localization result successfully. Our algorithm can support an output refreshing every 5 s with a normal desktop. Based on a literature research, this was the first time a real-time microseismic localization result was acquired. Although the refresh rate is not as high as in other lightweight IoT applications, it is a great improvement over geophysical exploration approach.

The offline experiments showed the robustness of our algorithm. When signal is noisy and packet loss is occurring in the network, the proposed algorithm still works, and the error does not increase too much. The time synchronization experiment result showed that the algorithm can work under small time shifts (less than 5 ms) between different nodes. The undulating surface test showed that the algorithm works well even if the sensor line is not horizontal. To sum up, the proposed algorithm is robust when it is implemented in the distributed sensor network, and it is ready for launch on a real hardware-based application.

## 6. Conclusions and Future Work

We designed a distributed sensor network system for underground seismic source localization. We have a simulation platform to test our algorithm; the connected ports for real-time data from real devices and the visualization system are also ready for future experiments. We used an optimized GB-TRI algorithm to rapidly calculate passive source localization in a distributed and decentralized way. The GB-TRI algorithm has good performance when the algorithm is implemented in a large-scale distributed network. The real-time and one-time experimental results showed that our proposed system can provide a real-time localization result with a 5 s refresh period; and the optimized GB-TRI algorithm is robust to noisy signals, packet loss, poor time synchronization and topographic variation.

The current GB-TRI method still has some shortcomings. Although a Gaussian beam simplifies the wavefield solution, we still need to set up the whole velocity model as the input to every node. The algorithm still requires the wavefield data to be exchanged between nodes. If a spatially distributed wavefield solution is available, the communication cost will be further reduced.

Our future plan is to improve the imaging algorithm to make it more efficient, and also test the real-time performance in a distributed network. We are going to embed our algorithm in a real device to test whether the algorithm works well in a small embedded system. We also plan to modify the structure of the algorithm to adjust to real-device deployment. In general, altering the algorithm will be the very first stage of this work; there is still a lot of work to do regarding the engineering. To perform real-time experiments with real devices, we still need to deal with detailed problems. 

## Figures and Tables

**Figure 1 sensors-21-02830-f001:**
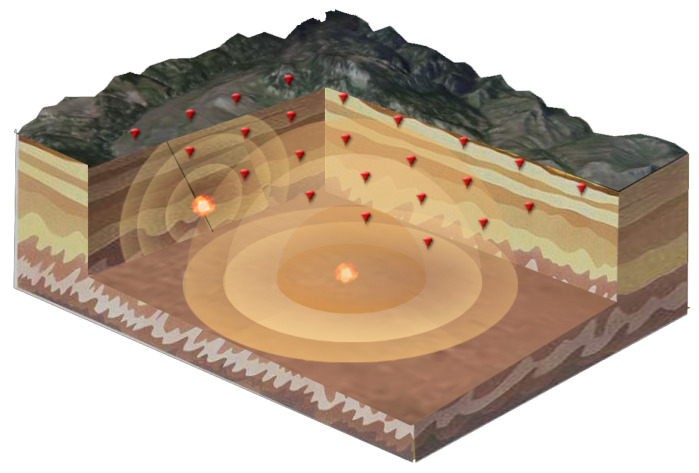
An illustration of the seismic sensor network for microseismic source event localization.

**Figure 2 sensors-21-02830-f002:**
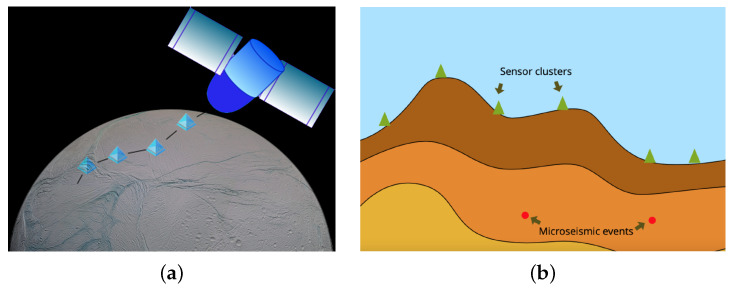
(**a**) Dropping the cabled sensor clusters from a low-altitude aircraft. (**b**) The 2D model for seismic event localization with topographic variation.

**Figure 3 sensors-21-02830-f003:**
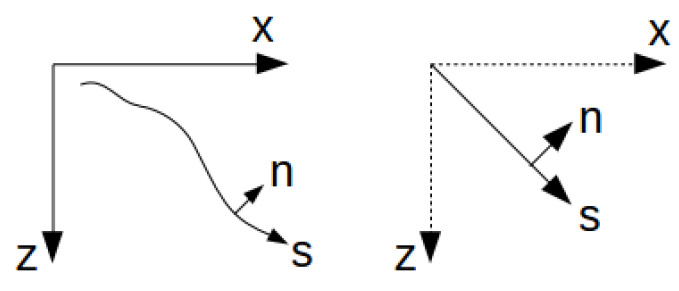
Ray-center coordinates and ray-direction coordinates.

**Figure 4 sensors-21-02830-f004:**
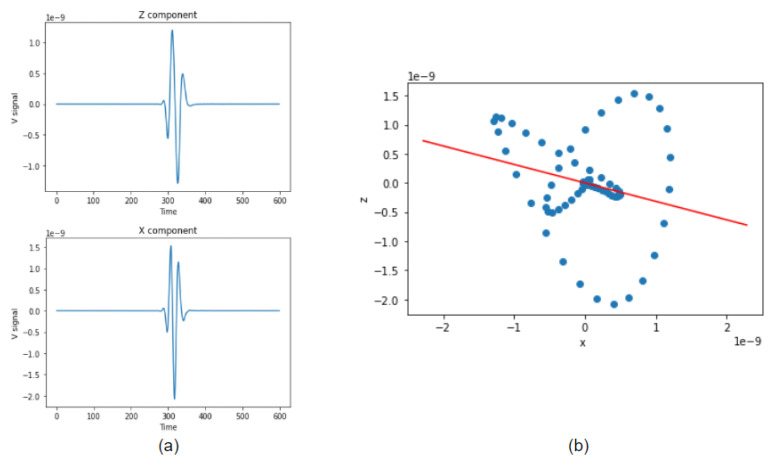
(**a**) Seismic source recorded on two different channels. (**b**) Wave propagation direction detected by SVD.

**Figure 5 sensors-21-02830-f005:**
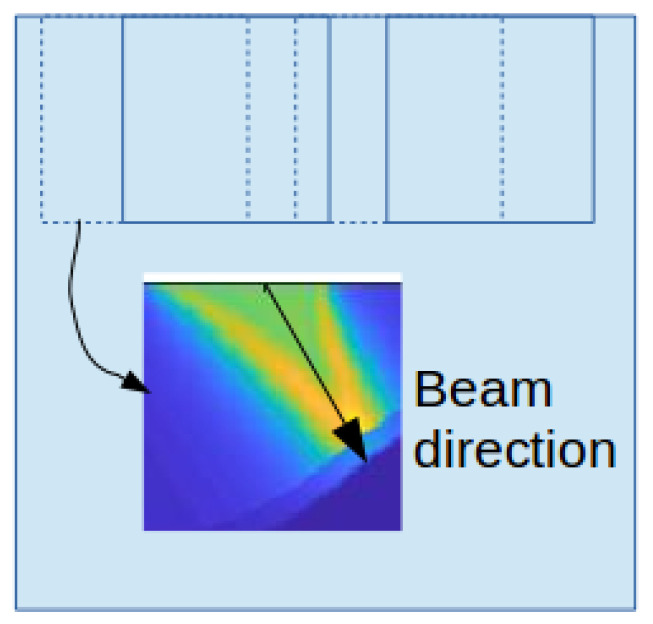
Calculating the direction of Gaussian beam with the superficial layer TRI image.

**Figure 6 sensors-21-02830-f006:**
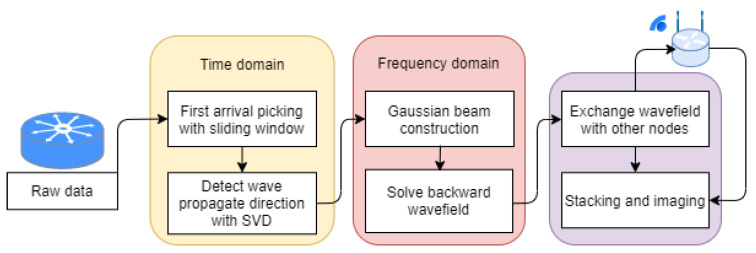
Workflow for source localization on a node in a distributed sensor network system.

**Figure 7 sensors-21-02830-f007:**
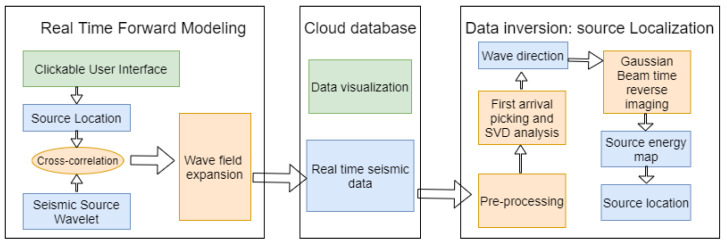
The framework of the real-time microseismic generation and inversion platform.

**Figure 8 sensors-21-02830-f008:**
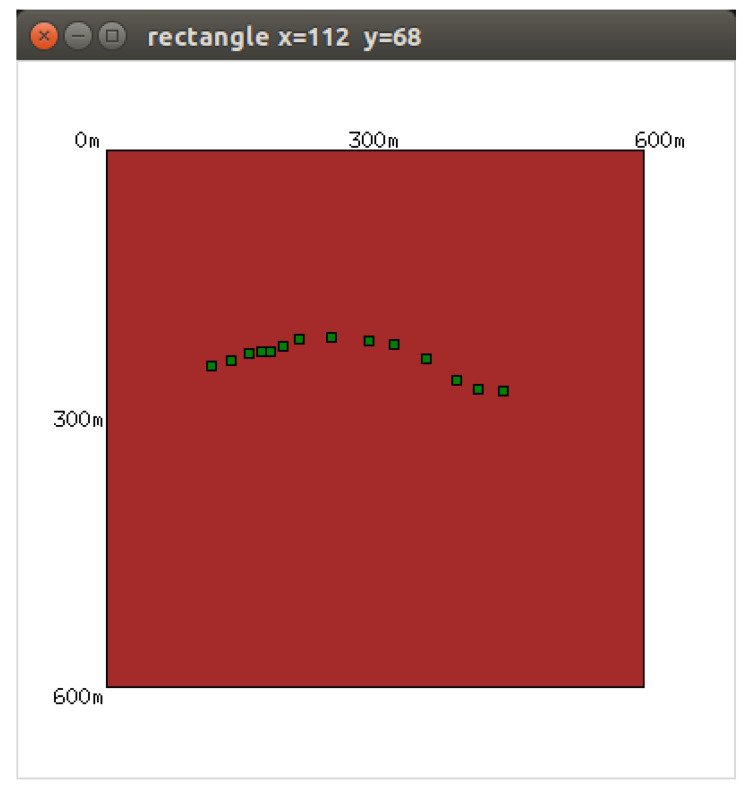
Clickable GUI for generating real-time seismic vibration data.

**Figure 9 sensors-21-02830-f009:**
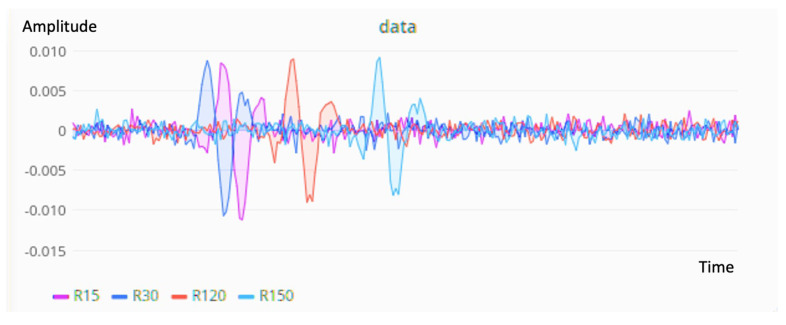
Real-time visualization of seismic data recorded on different nodes.

**Figure 10 sensors-21-02830-f010:**
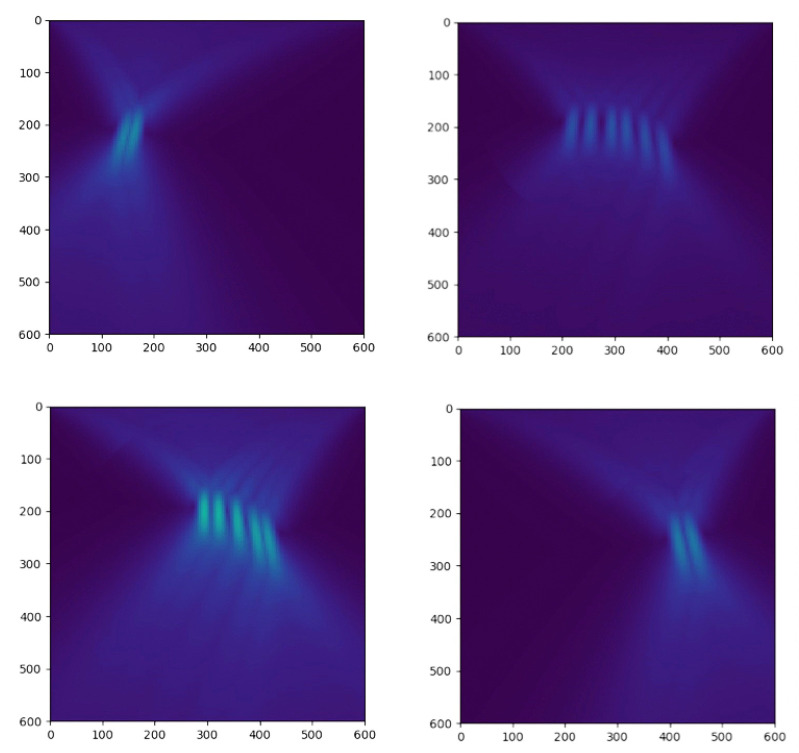
The screenshots of the output energy map.

**Figure 11 sensors-21-02830-f011:**
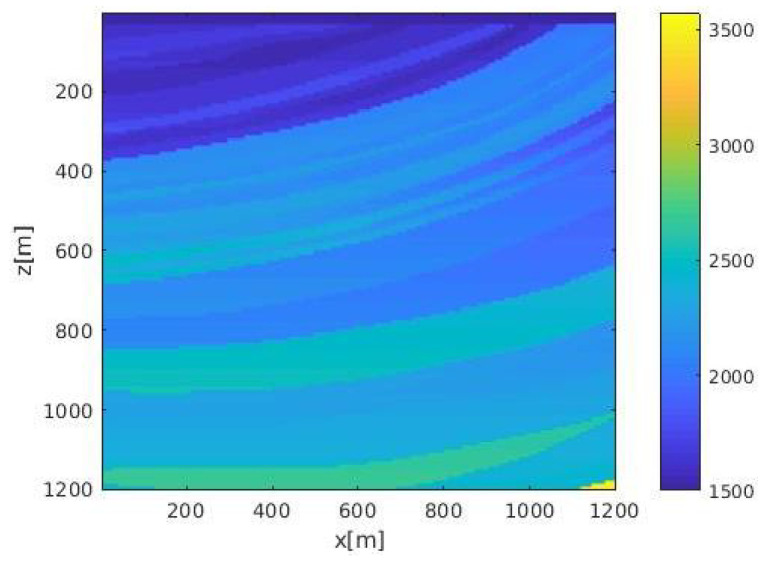
Marmousi velocity model.

**Figure 12 sensors-21-02830-f012:**
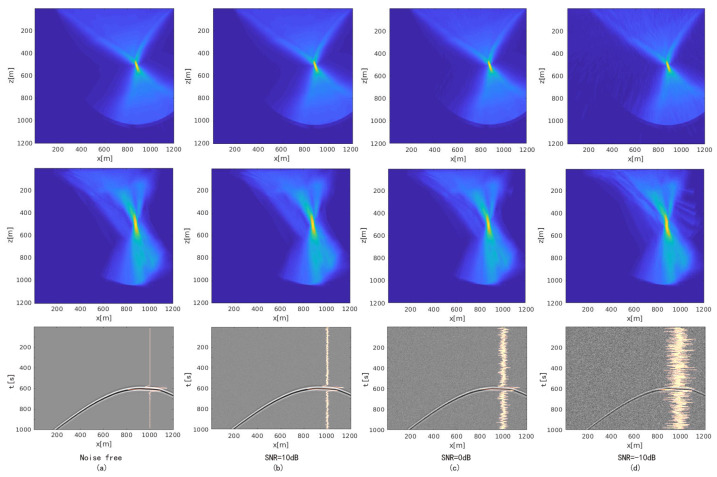
The FDTD-TRI image (**first row**), the GB-TRI image (**second row**) and the observed data (**third row**) with different SNRs. (**a**) Noise free; (**b**) SNR = 10 dB; (**c**) SNR = 0 dB;(**d**) SNR = −10 dB.

**Figure 13 sensors-21-02830-f013:**
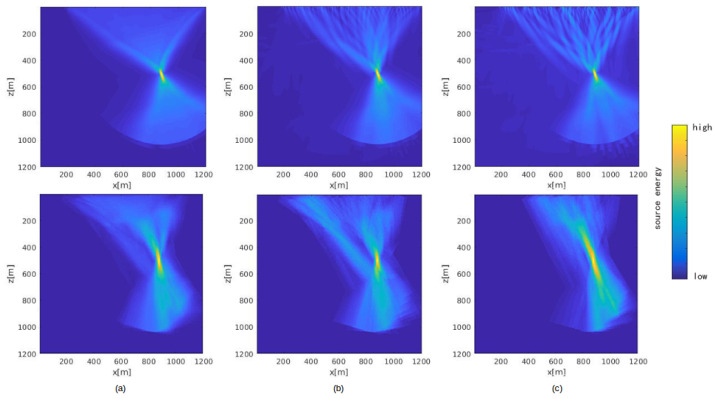
FDTD-TRI (**first row**) and GB-TRI (**second row**) energy maps under different packet loss rates: (**a**) 0% packet loss; (**b**) 40% packet loss; (**c**) 70% packet loss.

**Figure 14 sensors-21-02830-f014:**
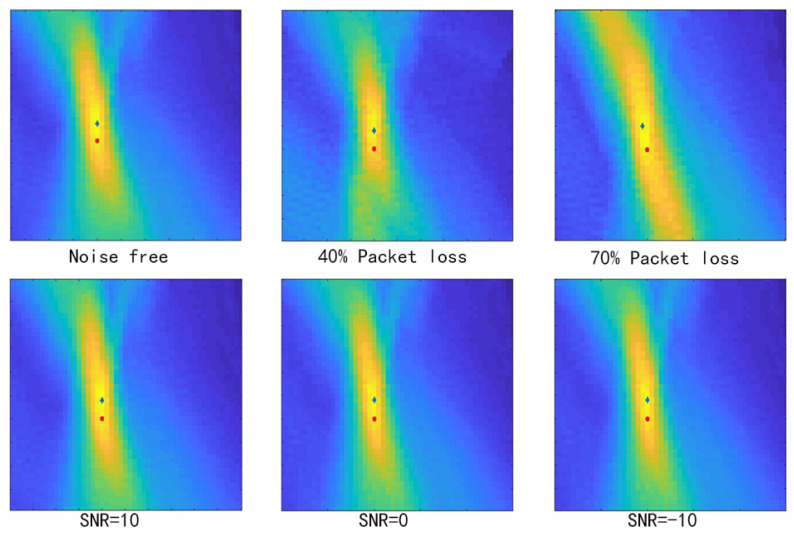
The comparison between the location result calculated by the GB-TRI method and the true location of the seismic source under different noise and packet loss conditions.

**Figure 15 sensors-21-02830-f015:**
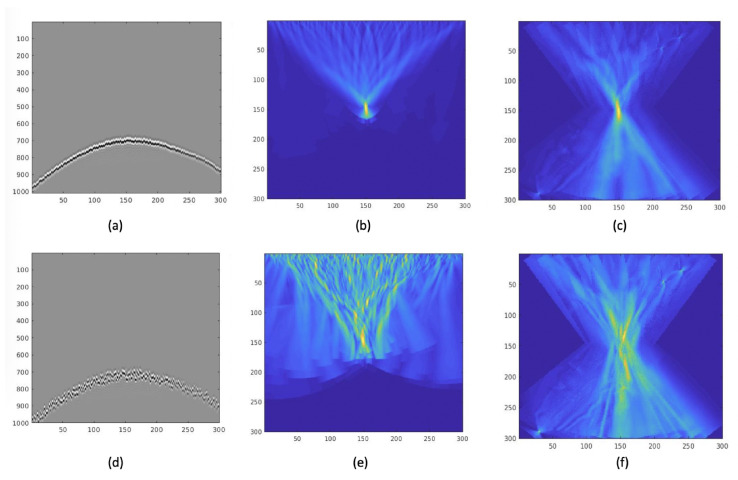
(**a**) Seismic data with random delays near 5 ms; (**b**) FDTD-TRI result; (**c**) GB-TRI result; (**d**) seismic data with five times larger random delays; (**e**) FDTD-TRI result; (**f**) GB-TRI result.

**Figure 16 sensors-21-02830-f016:**
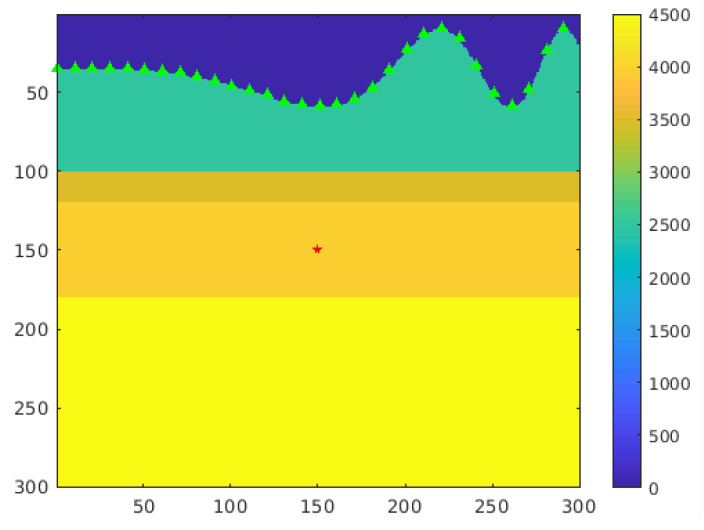
Velocity model with topographic variation. The green triangles are sensors, and the red star is the seismic source.

**Figure 17 sensors-21-02830-f017:**
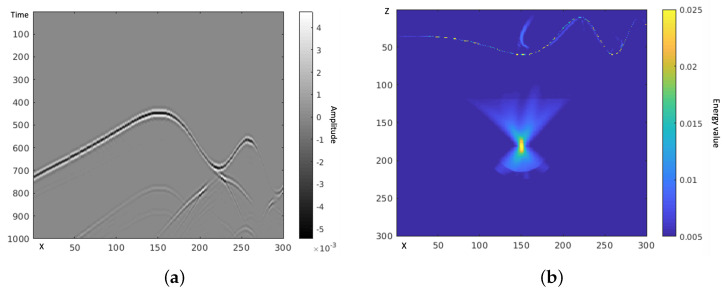
(**a**) The record of the observed seismic data; (**b**) the localization result.

**Table 1 sensors-21-02830-t001:** A comparison of different microseismic localization methods.

Method	Computation Device	Data Transmission	Time Consuming
Traditional TRI	computer cluster	hardware transportation	several months
Distributed TRI	smart sensor	internet	several minutes
GB-TRI	smart sensor	internet	several seconds

**Table 2 sensors-21-02830-t002:** Communication and computational cost analysis.

Algorithm	Communication	Computation	Computation
	Cost	Cost	Cost Per Node
Centralized TRI	t×l×m	$o({\left(z\times x\right)}^{2})$	$o({\left(z\times x\right)}^{2})$
Distributed TRI	z×x×t×l×m	$o(l\times {\left(z\times x\right)}^{2})$	$o({\left(z\times x\right)}^{2})$
GB-TRI	z×x×m	o(l×z×x)	o(z×x)

**Table 3 sensors-21-02830-t003:** Deviation under different SNR and packet loss rate.

Deviation (m)	SNR = 10 dB	SNR = 0 dB	SNR = −10 dB
0% packet loss	17.9815	20.6491	27.2628
40% packet loss	17.4121	24.3091	27.2861
70% packet loss	31.7856	103.5200	153.9617

## Data Availability

Some of the experimental data were from the SEG online public library.

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
