# Peer review of "Underground Microseismic Event Monitoring and Localization within Sensor Networks"

_sensors, 2021, doi:10.3390/s21082830_

Round 1

Reviewer 1 Report

Dear Authors,

Thank You for the opportunity of reading this article. My general opinion about the article is positive.

General statements about the article:

-> article proposes  a decentralized GB-TRI) algorithm that can be incorporated to the distributed sensors to detect and locate underground microseismic events with reduced the usage of computation resources and communication bandwidth of the network. So the area of the research is actual and highly desirable.

-> The article content suite to Sensors journal scope

-> the organization of the article is correct. Section are ordered in correct manner.

-> abstract is adequate to article content

-> Keywords are correctly proposed.

-> Methods are clearly introduced. All essential information is introduced.

-> Literature review is based on 34. They are related to article content.

-> quality of the presentation is high.

However, I indicate some issues that require additional clarification:

#1

The results in the article are well presented. This is a strong element of the article but there is a lack of discussion. Thus please add a separate section with discussion.

#2

Please add a paragraph at the end of the introduction section with the article organization. I mean “Section 2 presents …, section 3 concerns …, etc”

#3

Please also revise the manuscript regarding the personal way of addressing in the text. Please avoid and replace we" or "our" with the impersonal manner of addressing. The text will sound much more professional.

Technical issues:

-> Please add data availability statement as well as authors contribution.

-> please use bullet list in lines 47- … to highlight presented contribution

Author Response

Dear Reviewer:

Thank you for taking the time to review our paper. We made revision base on your comments:

#1

Thanks for your advice! We add section 5 for discussion. The experiment result is discussed, and the innovation of this work is addressed.

#2

Thanks for the comments! I add the paragraph at the end of the introduction section to introduce the article organization.

#3

Thank you for your advice. I asked a native speaker to help me go over the text, and revised the writing. I hope that the writing of this paper is satisfactory now.

Technical issues:

-> Please add data availability statements as well as authors contribution.

Answer: I added it in acknowledgment. 

-> please use bullet list in lines 47- … to highlight presented contribution

Answer: Thanks for the advice, I rewrote that paragraph.

Reviewer 2 Report

The article presents a decentralized Gaussian Beam Time Reverse Imaging (GB-TRI) algorithm that can be incorporated to the distributed sensors to detect and locate underground microseismic events. The results obtained are good, but the writing of the paper can further be improved. However, the following issues should be addressed:

  • Introduction section should include more works from recent past.
  • The last paragraph “The contribution of our work….” at the end of the Introduction section, should clearly highlight the novelty of your proposed work. How your work outperforms other recent works.
  • It would be clear if the authors compare the proposed work with more similar works reported in recent past in a Table form.
  • The shortcomings of the project should also be included with reasons in Conclusion and Future work section.
  • The English writing of the paper can further be polished. It needs a thorough review from an English expert.

Author Response

Dear Reviewer:

Thank you for taking the time to review our paper. We made revision base on your comments:

#1

The introduction section should include more works from the recent past.

Response:

Thanks for your advice! We added the related work from 2015 to 2019 to the introduction. 3 additional references are added. Also, I modify section 2 related a little bit to add more references.

#2

The last paragraph “The contribution of our work….” at the end of the Introduction section, should clearly highlight the novelty of your proposed work. How your work outperforms other recent works.

Response:

Thanks for your advice! I added the words to address our innovation. Also, we use a bullet list to highlight the contribution. 

#3

It would be clear if the authors compare the proposed work with more similar works reported in the recent past in a Table form.

Response:

Thanks for your advice! I added a new table in section 2 to compare our approach to the traditional approach and the previous distribution approach. Some discussion is also added to section2.

#4

The shortcomings of the project should also be included with reasons in Conclusion and Future work section.

Response:

Thanks for your comment! We added a new paragraph in the Conclusion and Future work section to discuss the shortcomings.

#5

The English writing of the paper can further be polished. It needs a thorough review from an English expert.

Response:

Thank you for your advice. A native speaker helped me go over the text. And also I went to the writing center for proofreading. I hope that the writing of this paper is satisfactory now.

Reviewer 3 Report

As per my review, I have noticed some minor flaws only ( as i mentioned for Authors already, in the Authors comment column  of the review report)
As per me, that paper " Underground Microseismic Event Monitoring and Localization within Sensor Networks" has written well by authors and also well organized. Regarding some points that need to be addressed by authors, are as follows:
1. Keywords, need to be added few more key terms
2.Figures 1, and 2 missing references
3. Figure 7, the title should be shortened and clear
4.For equations 6, 7, and 8, missing abbreviations for variables used in equations
5. Some of the key terms, missing abbreviation (like AWS page2, line no. 45)

Note: As per my review, the literature part in Introduction, (contributions and research gap, all are satisfied, and presented up to the level of standard), Results and discussion, Conclusion part also well demonstrated. Thatswhy, i have given minor points to address in my review report form.

Author Response

Dear Reviewer:

Thank you for taking time to review our paper. We made revision base on your comments:

#1

 Keywords, need to be added few more key terms

Response:

Thank you for your advice. We modified the existing keywords and added more keywords.

#2

Figures 1, and 2 missing references

Response:

 Thanks for comments. Both figure 1 and figure 2 are drawn by myself. So we don’t have the reference. Figure 1 has been used in one of my previous published paper, I added the citation of that paper. Figure 2 are new pictures I draw for this paper.

#3

 Figure 7, the title should be shortened and clear

Response:

 Thanks for comments. I modified the title of the figure7.

#4

For equations 6, 7, and 8, missing abbreviations for variables used in equations

Response:

 Thanks for comments. The  variables used in equations 6, 7, 8 are explained in the paragraph following the equation. I modified those three paragraphs to make them more clear.

#5

Some of the key terms, missing abbreviation (like AWS page2, line no. 45)

Response:

 Thanks for comments. I go through the whole paper again and fix these problems.

Round 2

Reviewer 2 Report

The revisions made are satisfactory.